# Modulation of the Inflammatory Response in Polycystic Ovary Syndrome (PCOS)—Searching for Epigenetic Factors

**DOI:** 10.3390/ijms232314663

**Published:** 2022-11-24

**Authors:** Dariusz Szukiewicz, Seweryn Trojanowski, Anna Kociszewska, Grzegorz Szewczyk

**Affiliations:** 1Department of Biophysics, Physiology & Pathophysiology, Faculty of Health Sciences, Medical University of Warsaw, 02-004 Warsaw, Poland; 2Chair and Department of Obstetrics, Gynecology and Gynecological Oncology, Medical University of Warsaw, 03-242 Warsaw, Poland

**Keywords:** polycystic ovary syndrome, epigenetic factors, inflammatory response, oxidative stress, mitochondrial dysfunction, insulin resistance, androgenic activity modulation

## Abstract

Polycystic ovary syndrome (PCOS) is the most common endocrine disorder in women of reproductive age. Despite its incidence, the syndrome is poorly understood and remains underdiagnosed, and female patients are diagnosed with a delay. The heterogenous nature of this complex disorder results from the combined occurrence of genetic, environmental, endocrine, and behavioral factors. Primary clinical manifestations of PCOS are derived from the excess of androgens (anovulation, polycystic ovary morphology, lack of or scanty, irregular menstrual periods, acne and hirsutism), whereas the secondary manifestations include multiple metabolic, cardiovascular, and psychological disorders. Dietary and lifestyle factors play important roles in the development and course of PCOS, which suggests strong epigenetic and environmental influences. Many studies have shown a strong association between PCOS and chronic, low-grade inflammation both in the ovarian tissue and throughout the body. In the vast majority of PCOS patients, elevated values of inflammatory markers or their gene markers have been reported. Development of the vicious cycle of the chronic inflammatory state in PCOS is additionally stimulated by hyperinsulinemia and obesity. Changes in DNA methylation, histone acetylation and noncoding RNA levels are presented in this review in the context of oxidative stress, reactive oxygen species, and inflammatory signaling in PCOS. Epigenetic modulation of androgenic activity in response to inflammatory signaling is also discussed.

## 1. Introduction

Polycystic ovary syndrome (PCOS) is one of the most common hormonal disorders among women of reproductive age, with a global prevalence ranging from 4% to 20% [1,2,3]. Such a large spread in the reported PCOS frequency reflects the heterogenous nature of this complex disorder that merges genetic, environmental, endocrine, and behavioral factors. Primary clinical manifestations of PCOS are derived from the excess of androgens and include absence of ovulation (anovulation), polycystic ovarian morphology, lack of or scanty, irregular menstrual bleedings, acne—an inflammatory condition of the skin in which the skin’s sebaceous glands become clogged and infected, and excessive growth of dark or coarse hair in a male-like pattern—face, chest and back (hirsutism). The secondary manifestations include multiple metabolic, cardiovascular, and psychological disorders (Figure 1A). The syndrome is poorly understood and remains underdiagnosed, and female patients are diagnosed with a delay [4]. This creates serious problems, as the complexity of PCOS requires a quick diagnosis and the development of therapeutic strategies for long-term health issues known as PCOS-related complications, such as insulin resistance (IR) among the metabolic abnormalities. The course of PCOS characteristically evolves with age, from a reproductive disease to a more metabolic disorder with increased incidence of type 2 diabetes and cardiovascular disease (e.g., atherosclerosis, high blood pressure) in later life [4,5].

Accumulating evidence suggests a multifactorial etiology of PCOS, with different genetic variants playing crucial roles in the pathogenesis. However, dietary and lifestyle factors also play important roles in the development and course of PCOS, suggesting strong epigenetic and environmental influences [7].

The phenotypes of PCOS can be subdivided into four different types as included in the simple clinical classification (Figure 1B). Although PCOS occurs regardless of the body mass index (BMI) value, overweight or obesity coexists in at least 60% of patients, and the prevalence and severity of clinical features in PCOS are positively correlated with increased BMI and hyperinsulinemia [8]. Moreover, an increased prevalence of nonalcoholic fatty liver disease (NAFLD) and obstructive sleep apnea has been reported in patients with PCOS, especially in those with features of metabolic syndrome. Obesity, in particular central adiposity, IR, a chronic proinflammatory environment, and excess androgens are considered the main factors related to NAFLD in PCOS [9,10,11]. Although IR is not one of the Rotterdam diagnostic criteria [i.e., the presence of two of three of the following criteria: oligo-anovulation, hyperandrogenism and polycystic ovaries (≥12 follicles measuring 2–9 mm in diameter and/or an ovarian volume > 10 mL in at least one ovary)], it has been reported in up to 80% of obese PCOS patients and 20–25% of lean PCOS patients [6,12].

Many studies have shown a strong association between PCOS and chronic, low-grade inflammation both in the ovarian tissue and throughout the body [13,14,15]. Indeed, in the vast majority of PCOS patients, elevated values of inflammatory markers or their gene markers have been reported. Constantly updated comparative studies (PCOS women vs. PCOS-free age- and BMI-matched controls) have revealed typical components of the inflammatory background in PCOS that include C-reactive protein (CRP) and high-sensitivity CRP (hs-CRP), interleukin 18 (IL-18), tumor necrosis factor alpha (TNF-α), interleukin 6 (IL-6), white blood cell count (WBC), monocyte chemoattractant protein-1 (MCP-1), complement element 3 (C3), and macrophage inflammatory protein-1α (MIP-1α) [14,16]. Interestingly, the association between diet and PCOS might be mediated by the inflammatory properties of the diet. It has been recently demonstrated that a high dietary inflammatory index was associated with increased odds of PCOS diagnosis [17].

Advanced glycation end products (AGEs) or “glycotoxins” are another complex group of heterogeneous compounds whose concentrations are increased in women with PCOS [18]. AGEs are generated mainly in the late stages of the Maillard reaction, which occurs when reducing sugars react in a nonenzymatic manner with amino acids in proteins, lipids or DNA [19]. The formation of AGEs is usually endogenous but can be derived from exogenous sources such as tobacco smoke or diets containing high levels of AGEs (e.g., fast-food diets) [20,21]. Acting via their cellular membrane receptor RAGE or using RAGE-independent pathways, AGEs are involved in the pathogenesis of aging and aging-related or degenerative diseases such as atherosclerosis, Alzheimer’s disease, IR-related diabetes, or kidney disease [22]. What is most important here is that by causing oxidative stress, altering enzymatic activities, affecting cytotoxic pathways, or damaging nucleic acids, AGEs can contribute to the pathogenesis of PCOS as well as its consequences. It has been demonstrated that chronic inflammation and increased oxidative stress may be a link between the mechanisms of AGE action and the metabolic and reproductive consequences of PCOS [23,24]. Moreover, hyperandrogenism in PCOS induces endoplasmic reticulum (ER) stress in granulosa cells (GCs), resulting in increased expression of RAGE and accumulation of AGEs in the ovary [25]. Such proinflammatory AGE–RAGE signaling is seen as the cause of both altered steroidogenesis and folliculogenesis in PCOS [18,26].

Development of the vicious cycle of the chronic inflammatory state in PCOS is additionally stimulated by hyperinsulinemia and obesity [27]. There are also markers of endothelial dysfunction/damage in PCOS, including asymmetric dimethylarginine (ADMA), CRP, homocysteine, plasminogen activator inhibitor-I (PAI-I), PAI-I activity, and vascular endothelial growth factors (VEGFs) [14,27,28]. Endothelial dysfunction and an altered cytokine profile toward proinflammatory and prothrombotic conditions in PCOS may increase cardiovascular risk and limit reproductive capacity to a large extent (ranging from anovulation through miscarriage to placental insufficiency) [28,29,30,31]. Additionally, treatment of ovulatory dysfunction is hampered by the fact that abnormal TNF-α, angiopoietin-2, and adiponectin levels may be associated with clomiphene resistance in PCOS patients. Therefore, the serum proinflammatory cytokines TNF-α and adiponectin are helpful discriminators of clomiphene resistance in PCOS patients [32,33].

In recent years, the participation of epigenetic mechanisms in the etiopathogenesis of PCOS has been a problem attracting increasing attention from researchers [34,35,36]. It has been shown beyond reasonable doubt that epigenetic factors are able to initiate and control the inflammatory response and modulate metabolic changes in PCOS, including IR [37,38,39]. Importantly, from a therapeutic point of view, unlike genetic changes (e.g., mutations), epigenetic regulation may be reversible because it does not influence DNA sequences [40,41]. Epigenetic mechanisms provide an adaptive margin of control in the modulation of gene expression. In this way, the whole body (as well as individual systems and organs) can adapt to a changing environment. Moreover, epigenetic modifications regulate development in eukaryotic organisms and may promote the effects of parental origin by genomic imprinting. Thus, genomic imprinting is an epigenetic phenomenon that causes genes to be expressed or not, depending on whether they are inherited from the mother or the father [42].

Gene expression may be significantly influenced by the increased functional complexity of DNA resulting from altered chromatin structure, nuclear organization, and transcript stability. Relevant epigenetic mechanisms include DNA methylation, histone modification, protein phosphorylation, noncoding RNAs (ncRNAs), and RNA modifications (i.e., methylation, editing and splicing). For example, epigenetic regulation by miRNA targets approximately 60% of human genes [43]. An overview of the main epigenetic mechanisms is presented in Figure 2.

Changes in DNA methylation, histone acetylation and noncoding RNA levels have been demonstrated in PCOS. It is likely that altered DNA methylation may predispose to the deregulation of genes involved in the inflammatory response, hormone production and signaling as well as glucose and lipid metabolism [44]. Based on a recently published meta-analysis, it can be concluded that, compared to PCOS-free controls, PCOS patients show significant global DNA hypomethylation in different tissues and peripheral blood. A more detailed assessment focused on single-gene methylation revealed that genes associated with several functions were significantly hypo- and hypermethylated in patients with PCOS [45]. It was established, inter alia, that increased expression of the anti-Müllerian hormone receptor (AMHR) gene in ovarian tissue and decreased expression of the insulin receptor (IR) gene in the endometrium of PCOS women were correlated with decreased and increased methylation levels of these genes, respectively [46].

The aim of this review is to present the state of the art regarding the involvement of epigenetics in the regulation of the inflammatory response and in PCOS.

### Methodology of Literature Searching

Scopus^®^ and Pubmed^®^ databases have been reviewed using the following keywords in the context of polycystic ovary syndrome (PCOS): epigenetic factors, inflammatory response, oxidative stress, mitochondrial dysfunction, insulin resistance, androgenic activity modulation. Briefly, this procedure was aimed at collecting evidence on the inflammatory processes that may influence the epigenetic regulation of genes and thus alter the gene expression in women with PCOS. The respective original research papers, comprehensive reviews and meta-analyses were retrieved. To avoid excessive number of citations, highly cited original research papers of recognized authors and most recent high quality reviews by experts in the field were preferred.

## 2. Oxidative Stress and Inflammatory Signaling in PCOS

Oxidative stress is the result of long persistence of the imbalance between the production of reactive oxygen species (ROS) and their elimination by protective mechanisms [47]. ROS include both free radical and nonfree radical oxygenated molecules, such as hydrogen peroxide (H_2_O_2_), superoxide (O_2_^−^), singlet oxygen (1/2 O_2_), and the hydroxyl radical (∙OH). Oxidative stress can also be triggered by the accumulation of reactive nitrogen, iron, copper, and sulfur species [48,49]. Free radicals are species possessing unpaired electrons in the external orbit that can exist independently [49,50].

Independent studies have shown that oxidative stress is inherent in PCOS, causing altered steroidogenesis in the ovaries, which subsequently contributes to increasing androgen levels, disturbing follicular development and resulting in infertility [51,52]. Indeed, the concentrations of oxidative stress markers such as malondialdehyde (MDA), advanced glycosylated end products (AGEs), nitric oxide (NO), and xanthine oxidase (XO) are typically increased in the serum of PCOS patients [52]. The role of oxidative stress in the pathophysiology of PCOS may be because ROS—acting as main regulators of inflammatory signaling, particularly with respect to nuclear factor kappa-light-chain-enhancer of activated B cells (NF-κB) activation and inflammasome signaling—may perpetuate the inflammatory background [13,47]. Moreover, IR in PCOS is significantly related to the total oxidant status (TOS) and the level of inflammatory factor activity [53].

### 2.1. Epigenetic Landscape Related to ROS Formation in PCOS (See Also Figure 3, Which Corresponds to the Text of this Section)

One of the best-characterized epigenetic modifications of chromatin is DNA methylation. Homocysteine (Hcy) is an intermediate product of one-carbon metabolism that donates methyl groups for methylation processes involved in epigenetic gene regulation [54]. In addition, excess Hcy in blood (hyperhomocysteinemia) promotes oxidative stress in vascular endothelial cells and surrounding tissues via increased formation of ROS [55]. In other words, Hcy-induced cell injury occurs because the autooxidation process of Hcy is highly reactive at physiological pH and leads to the production of superoxide and hydrogen peroxide [55,56,57]. Independent studies have demonstrated that compared with PCOS-free controls, increased serum Hcy levels are significantly associated with inflammatory response, androgen excess, IR, hyperinsulinemia, and BMI in women with PCOS [58,59,60,61]. Compared with the normal Hcy group, patients with hyperhomocysteinemia had increased numbers of CD14^++^CD16^+^ inflammatory monocytes in their peripheral blood and elevated plasma levels of interleukin-1β and interleukin-6, the two typical cytokines secreted by inflammatory monocytes [59]. As demonstrated in both laboratory and clinical settings, deregulation of one-carbon metabolism and hypermethylation of mitochondrial DNA in the case of elevated Hcy concentrations in serum and follicular fluid may be involved in poor oocyte quality in PCOS [62,63,64].

**Figure 3 ijms-23-14663-f003:**
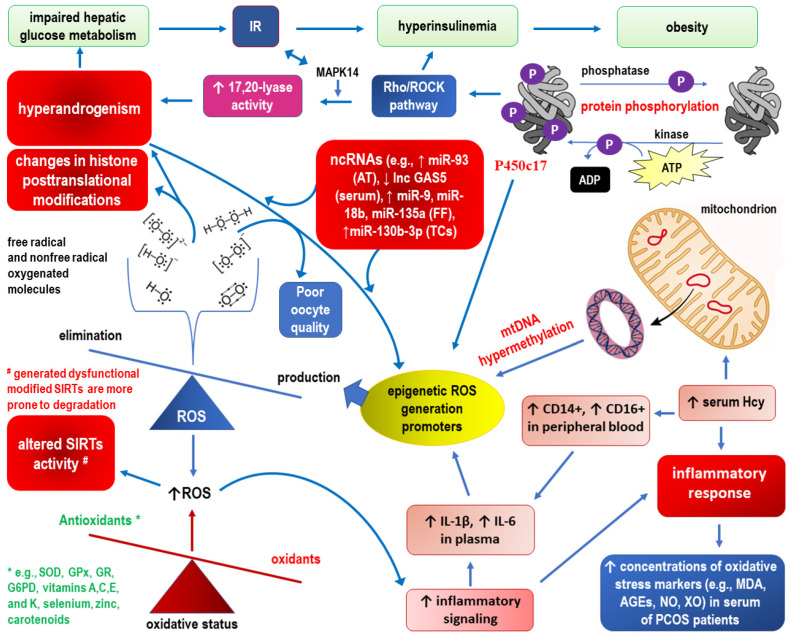
Epigenetic landscape related to ROS formation in PCOS. See the main text (Section 2.1) for detailed explanations.

In addition to hyperhomocysteinemia, other common cardiovascular risk factors promote the generation of ROS, including hyperlipidemia/dyslipidemia, glucose intolerance, IR and diabetes, metabolic syndrome, obesity, hypertension, aging, and—theoretically more controllable—smoking, alcohol, and unhealthy diet [65]. Substantially, all these cardiovascular risk factors are reported in PCOS, indicating that the epigenetic landscape for ROS formation is varied and runs in parallel with cardiovascular threats [66].

In addition to methylation, as in the case of hyperhomocysteinemia, the entire spectrum of epigenetic mechanisms is involved in the pathomechanisms of the mentioned “ROS generation promoters”. For example, noncoding RNAs are involved in IR, such as increased expression of miR-93 in adipose tissue and decreased expression of lncRNA GAS5 in the serum [67,68]. Moreover, increased expression levels of miR-9, miR-18b, and miR-135a in follicular fluid promote steroid synthesis, whereas overexpression of miR-130b-3p in thecal cells is correlated with excessive androgen biosynthesis [69,70]. Along with subsequent research, an increasing number of aberrant long noncoding RNAs that increase oxidative stress have been detected in different specimens from women with PCOS (i.e., serum, follicular fluid, granulosa cells, cumulus cells) and in various PCOS rodent models (i.e., dehydroepiandrosterone (DHEA) and letrozole-induced models) [71]. Posttranslational regulation of androgen biosynthesis may occur at the level of protein phosphorylation. It was proposed that serine phosphorylation of the single enzyme cytochrome P450c17 (P450c17) via the Rho/Rho-associated coiled-coil containing protein kinase (Rho/ROCK) pathway increases 17,20-lyase activity [72]. Considering that P450c17 catalyzes both 17α-hydroxylation and 17,20-lyase conversion of 21-carbon steroids to 19-carbon precursors of sex steroids, the ratio of 17,20-lyase to steroid 17α-hydroxylase activity will change. Increased activity of 17,20-lyase following serine phosphorylation of P450c17 is a result of enhancement of the enzyme’s affinity for its redox partner, P450 oxidoreductase [72]. An overactivity of 17,20-lyase with a higher production of the testosterone precursors dehydroepiandrosterone (DHEA) and androstenedione/androstenediol seems especially present in hyperandrogenic PCOS [73]. The Rho/ROCK pathway uses mitogen-activated protein kinase 14 (MAPK14, p38α) as the kinase responsible for enhancing 17,20-lyase activity through P450c17 phosphorylation [74]. It has been demonstrated that MAPK14 signaling pathways are oxidant-sensitive and that oxidative stress exposure significantly induces dehydroepiandrosterone production through increased MAPK14 activation [75]. Considering that androgen excess in women impairs hepatic glucose metabolism by decreasing insulin-stimulated glucose uptake and glycogen synthesis, the MAPK14 pathway may be a nexus between IR and hyperandrogenism that is triggered by oxidative stress [75,76,77].

Free radicals generated during unbalanced/extended oxidative stress produce direct and indirect changes in histone posttranslational modifications. Histone H3 is more severely affected by oxidative stress-related processes. This may be because H3 is the only histone with cysteine residues, which can be glutathionylated and are redox-sensitive [78]. The activity of enzymes responsible for the demethylation and deacetylation of histones may be transiently altered by ROS [79]. For example, nicotinamide adenine dinucleotide NAD+-dependent histone deacetylase or sirtuin (SIRT) activity correlates with metabolic parameters of oxidative stress in PCOS [80]. A duality of observed responses has been demonstrated. Mild oxidative stress induces the expression of SIRTs probably as a compensatory mechanism, while severe or long-lasting oxidant conditions result in decreased SIRT levels due to the generation of dysfunctional modified SIRTs that are more prone to degradation by the proteasome [81]. In addition, SIRT3 deficiency in granulosa cells of PCOS patients may contribute to mitochondrial dysfunction, elevated oxidative stress, and defects in glucose metabolism, which potentially induce impaired oocytes in PCOS [82]. Therefore, the correlation between epigenetic factors and ROS generation can also manifest itself the other way around. It has also been suggested that increased oxidative stress and inflammation in PCOS may be due to decreases in antioxidants, such as peroxiredoxin 4 (Prx4) [53]. Interestingly, Prx4 is a related antioxidant in insulin synthesis and insulin signaling in adipose tissue [83,84]. However, to date, it has not been possible to unequivocally show a correlation between Prx4 levels and IR or carbohydrate status in PCOS patients [85].

Increasing evidence suggests that oxidative stress globally influences chromatin structure, DNA methylation, and enzymatic and nonenzymatic posttranslational modifications of histones and DNA-binding proteins. ROS not only induce an inflammatory response but also modulate the epigenetic landscape, which causes overproduction in PCOS. Thus, the vicious cycle may perpetuate the pathogenetic background of the disease [86].

### 2.2. Epigenetics of Mitochondrial Dysfunction in PCOS (See Also Figure 4, which Corresponds to the Text of this Section)

As a low-grade chronic inflammatory disease, PCOS patients have permanently elevated levels of inflammatory markers. Quite recently, interest was aroused by the notion that the proinflammatory background in PCOS, including the altered cytokine profile, may be induced by oxidative stress related to mitochondrial dysfunction [87]. Since mitochondria are known as guardians of the inflammatory response, it is plausible that mitochondrial epigenetics might play a pivotal role in inflammation [88]. Thus, inflammation and mitochondrial dysfunction have mutually destructive actions and induce a vicious cycle of functional deterioration. Another mechanism by which dysfunctional mitochondria evoke inflammatory responses involves the release of damage-associated molecular patterns (DAMPs) into the cytoplasm [89]. In other words, structural and functional disturbances of mitochondria can promote the production of inflammatory mediators, which in turn can further impair mitochondrial function. Even at a low intensity, long-lasting oxidative stress within the “energy centers of the cells” has negative effects on insulin resistance, lipid metabolism, and follicular development/ovulation [90].

**Figure 4 ijms-23-14663-f004:**
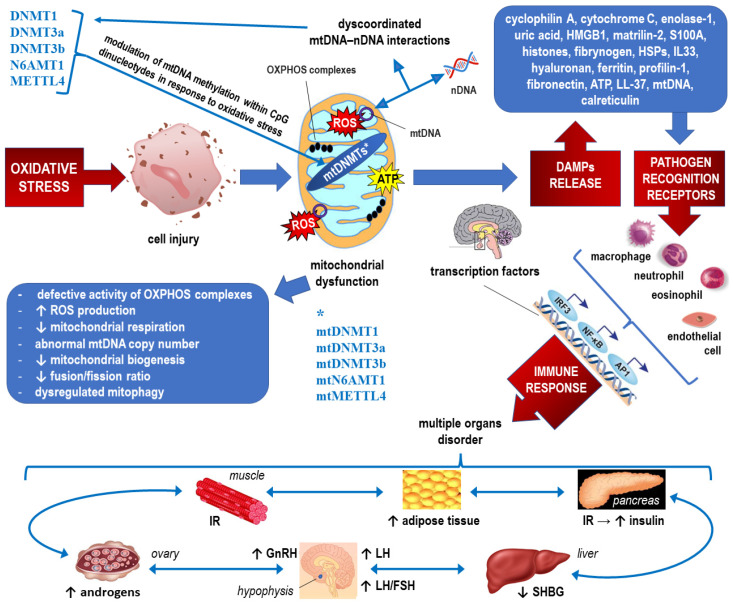
Epigenetics of mitochondrial dysfunction in PCOS. See the main text (Section 2.2) for detailed explanations. PCOS is caused by a vicious cycle of androgen excess, insulin resistance, low-grade inflammation, obesity and increased oxidative stress [91]. * Mitochondrial DNA methyltransferase 1, DNA methyltransferase 3 alpha, DNA methyltransferase 3 beta, N-6 adenine-specific DNA methyltrasferase 1, and methyltransferase 4, respectively.

Mitochondria are unique in that they contain their own circular DNA rather than linear DNA within the mitochondrial matrix, separate from the genomic nuclear DNA [92]. MtDNA encodes only 13 polypeptides, but the whole mitochondrial proteome comprises over 1500 proteins that are encoded by nuclear genes and translocated to the mitochondria for the purpose of maintaining mitochondrial function [93]. To ensure homeostasis, mitochondria have developed several mechanisms to sense and respond to stress via nuclear communication. It should be noted that precise control of mitochondria requires coordination between the two genomes: mitochondrial DNA (mtDNA) and nuclear DNA. Therefore, mitochondrial function can be affected by mutations and epigenetic modifications of nuclear gene expression, and whole-cell functionality may be modulated through mtDNA modifications [94,95].

Although there has been considerable debate regarding whether paternal mitochondrial DNA (mtDNA) transmission may coexist with maternal transmission of mtDNA, it is generally accepted that in humans, mitochondria and mtDNA are exclusively maternally inherited [96,97]. Although the structure of the mitochondrial genome is nonlinear but circular, three major epigenetic mechanisms that regulate gene expression within the mitochondrion include mtDNA methylation, noncoding RNAs, and posttranslational modifications of nucleoid-associated proteins. mtDNA molecules are arranged in clusters called nucleoids, which are tethered to the mitochondrial membrane and devoid of histones [98].

To date, the best-known epigenetic mechanism coupled to the adjustment of mitochondrial function is mtDNA methylation. During in vitro studies carried out in cell cultures under standard conditions, relatively low levels of mtDNA methylation were detected. This may be due to the structural arrangement of mtDNA within nucleoids (instead of tightly packed with histone proteins characteristic of nuclear DNA) and associated nucleoid proteins [99]. Moreover, in the mitochondria, methylation occurs within CpG dinucleotides, as CpG islands are virtually absent because of the smaller size of mtDNA and its short noncoding control region. In addition to cytosine methylation at position C-5 yielding 5-methylcytosine (5-mC), adenine methylation with the formation of 6-methyladenine (6-mA) has been confirmed in higher eukaryotes, including humans [100]. The respective DNA methyltransferases (DNMTs) responsible for cytosine methylation (DNMT1 for methylation maintenance and DNMT3a and DNMT3b for de novo methylation) are encoded in the nucleus and translocated to the mitochondria to form mtDNMTs [101]. Similarly, N-6 adenine-specific DNA methyltransferase 1 (N6AMT1) and methyltransferase like 4 (METTL4) are involved in adenine methylation [102]. It has been established that mtDNA transcription is accompanied by decreased in vivo methylation, whereas inducing mtDNA replication leads to increased methylation. However, the crucial phenomenon is the modulation of mtDNA methylation in response to oxidative stress because it may be clearly linked to mitochondrial dysfunction in PCOS correlated with elevated levels of inflammatory markers and a decrease in antioxidant capacity [99,103,104].

It has been proven, in vitro and in vivo, that increased de novo mtDNA methylation functions to protect the mitochondrial genome from mtDNA damage related to an increase in oxidative stress during embryogenesis [105]. However, accumulation of ROS with sublethal concentrations of H_2_O_2_ is capable of decreasing mtDNA methylation, indicating the possibility of an insufficient compensatory mechanism within mitochondria responsible for unblocking/activation of the antioxidant genes. However, it is also worth considering the possibility that nucleoids will remodel into a more compact state during oxidative stress to provide a more insulated environment for mtDNA [95,99]. In turn, persistent mitochondrial dysfunction in PCOS resulting in poor oocyte quality can manifest as impaired polar body extrusion and significantly decreased cleavage and blastocyst rates, which may be associated with mtDNA hypermethylation [62]. Such mtDNA hypermethylation in PCOS may be linked to hyperhomocysteinemia and subsequent significant upregulation of the one-carbon metabolic enzymes betaine homocysteine methyltransferase (BHMT) and glycine N-methyltransferase (GNMT) and the DNA methyltransferase DNMT1 [62]. Accordingly, with increased methyltransferase load, PCOS oocyte malfunction is reflected in hypermethylation of mtDNA sequences coding for 12S, 16S rRNA and ND4, as well as the initiating transcription D-loop region [62,63]. It may be presumed that the epigenetic protective mechanisms against mtDNA damage under oxidative stress may simultaneously inhibit the production of oocytes capable of being fertilized.

It has been demonstrated that mtDNMT1 expression in the mitochondrial matrix is upregulated by the transcription factors involved in mitochondrial biogenesis that activate the expression of nuclear-encoded mitochondrial genes in response to hypoxia and loss of p53, such as nuclear respiratory factor 1 (NRF-1), peroxisome proliferator activated receptor gamma coactivator-1 alpha (PGC-1α), and mitochondrial transcription factor A (TFAM) [101]. Thus, polymorphisms in the NRF-1, PGC-1α, and TFAM genes may be associated with different mitochondrial biogenesis in women with PCOS and therefore epigenetically modulate the pattern of response to oxidative stress [106,107].

Recent evidence indicates that noncoding RNAs regulate crosstalk between mitochondria and other cellular compartments. To date, the scant knowledge about the roles of mitochondrial noncoding RNAs (including nucleus-encoded noncoding RNAs that are translocated to the mitochondrion and mitochondrion-encoded noncoding RNAs that are released outside the mitochondria) in epigenetic modification of cellular processes in health and disease is constantly updated [108]. For example, gene ontology and pathway analysis showed that dysregulated lncRNAs in GCs in PCOS with hyperandrogenism have a regulatory role in mitochondrial function by interacting with transcription factors such as yin-yang 1 (YY1) and SIX homeobox 5 (SIX5) [109]. Corticotropin-releasing hormone-binding protein (CRHBP) is consistently the upregulated lncRNA with the highest fold-change in PCOS with hyperandrogenism compared to PCOS-free controls [109]. Importantly, mitochondrial oxidative function to maintain energy homeostasis in response to nutrient and hormonal signals is controlled by mammalian target of rapamycin (mTOR), a protein serine-threonine kinase acting via the YY1-PGC-1α transcriptional complex [110]. Considering that mTOR integrates signals from a variety of “energy-balancing” hormones such as leptin, insulin, and ghrelin, overexpression of YY1 resulting from dysregulated lncRNA may lead to oxidative stress, increased respiration and generation of ROS, with subsequent ATP depletion, opening of mitochondrial permeability transition, and apoptosis [110]. In addition to the inflammatory response due to disrupted mitochondrial biogenesis, some recent studies demonstrated that involvement of mTOR signaling or the PI3K/AKT/mTOR pathway is an important pathophysiological basis of PCOS because overexpression of the mTOR pathway can impair the interaction of cumulus cells, lead to IR, and directly affect follicle growth [111]. Research is ongoing to detect other mitochondrion-related miRNAs and lncRNAs involved in the epigenetic modulation of oxidative stress and the inflammatory response in PCOS [87,88,90].

Similarly, altered dynamics of the mitochondrial nucleoid or posttranscriptional modifications of more than 50 nucleoid-associated proteins may disturb the accessibility of various genes within mtDNA and predispose to the chronic inflammatory background in PCOS. However, the processes by which nucleoids are actively chosen for mtDNA replication and distribution within mitochondrial networks are still insufficiently explained. Oxidative stress may deteriorate the dynamics of nucleoids due to their resulting structural modifications and the breakdown of redox control that is evident in mitochondrial dysfunction. What is the effect and what is the cause is not yet clear [112,113].

## 3. Epigenetic Factors Influencing the Cytokine Profile and Inflammatory Markers in PCOS

Both the development and function of the immune system are strictly dependent on epigenetic pathways. Over the past decade, a significant number of publications on the importance of epigenetic modifications in the development of immune disorders have attracted the attention of researchers, including pathophysiologists involved in PCOS investigation [114,115].

As mentioned in the Introduction, there is no doubt that PCOS is associated with significant elevations of multiple markers of inflammation, including CRP, hs-CRP, IL-18, IL-34, TNF-α, TGFβ, IL-6, MCP-1, C3, and MIP-1α [14,16,116,117]. Furthermore, markers of endothelial dysfunction/injury and/or oxidative stress, such as ADMA, endothelin 1 (ET-1), PAI-I, visfatin, matrix metalloproteinase-9 (MMP-9), VEGFs, and soluble adhesion molecules ICAM-1 (Intercellular Adhesion Molecule 1 or CD54) and VCAM-1 (Vascular cell adhesion protein 1 or CD106), may be used to assess other derangements associated with inflammation in PCOS patients [14,52,118,119,120]. Considering that the proinflammatory action of AGEs can be neutralized with a sufficiently high concentration of the soluble form of the receptor for AGEs (sRAGE), sRAGE could represent another biomarker and a potential therapeutic target for ovarian dysfunction in PCOS [121]. Proinflammatory cytokines are produced predominantly by activated macrophages, indicating that epigenetic reprogramming of immune cells may play a crucial role in the development of an inflammatory background in PCOS. Indeed, women with PCOS had lower global DNA methylation in monocytes and macrophages that, acting at the level of cytosine-phosphate-guanine (CpG) dinucleotides of the CpG islands, may upregulate the expression of proinflammatory genes [122]. As mentioned elsewhere, hyperhomocysteinemia intensifies the inflammatory response in patients with PCOS by increasing the number of monocytes [59]. Moreover, reprogramming toward gene activation also applies to other immune cells that show lowered levels of DNA methylation, i.e., T helper, T cytotoxic, and B cells [115].

The results of the latest research show a significant impact of ncRNAs on the processes related to inflammation and immune response in PCOS. It was demonstrated that a total of 79 noncoding transcripts were differentially expressed in the GCs of PCOS patients [71,123]. For example, an 8.7 kb long ncRNA, long-chain noncoding RNA metastasis-related lung adenocarcinoma transcript 1 (MALAT1), has been found to interact with miRNAs in ovarian granulosa cells (GCs). MALAT1 regulates TGFβ signaling through binding with miR-125b and miR-203a. miR-125b directly targets transforming growth factor beta receptor 1 (TGFBR1), and miR-203a directly targets TGFBR2 [124]. The MALAT1 reduction typically observed in PCOS was identified to contribute to the repression of TGFβ signaling in GCs. Moreover, MALAT1 promoted the binding between the p53 protein and a primary cellular inhibitor of p53, the murine double minute 2 (MDM2) oncogene, which further boosted p53 proteasome-dependent degradation [125,126]. Leukemia inhibitory factor (LIF) activity may also be reduced by downregulation of MALAT1 in PCOS because this condition leads to the relative overexpression of inhibitory miR-302d-3p within the miR-302d-3p/LIF axis [127]. Considering that LIF is likely to coordinate follicular growth in the ovary, LIF deficiency may result in developing follicles undergoing atresia, presumably promoted by elevated androgen levels [128]. Therefore, it is assumed that MALAT1 plays a protective role in reducing ovarian tissue damage and endocrine disorders in PCOS accompanied by decreased levels of FSH and LIF and elevated serum levels of estrogen (E), testosterone (T), and luteinizing hormone (LH) by regulating the miR-302d-3p/LIF axis [127].

Continuing the topic of ncRNAs, an interaction between miR-19a and long ncRNA (lncRNA) placenta-specific protein 2 (PLAC2) may be crucial for the regulation of TNF-α signaling and consequently the activation of the transcription factor NF-κB and apoptosis in ovarian GCs [129]. As a multifunctional cytokine and adipokine, TNF-α exerts pleiotropic effects on many cell types and tissues, including ovarian and adipose tissues of PCOS patients [130]. miR-19a functions mainly by regulating TNF-α and therefore may play a critical role in the etiopathogenesis of PCOS. Inhibitory effects of miR-19a overexpression on the expression of TNF-α were demonstrated [131]. Thus, typically observed in PCOS, overexpression of miR-19a compared to controls may reflect, regardless of efficacy, the compensatory mechanism [132]. Interestingly, increased levels of TNF-α in PCOS were observed simultaneously with increased PLAC2 expression. It was documented that PLAC2 overexpression increases the expression levels of TNF-α by sponging miR-19a, thereby promoting apoptosis in granulosa cells. The latter is consistent with the well-established knowledge that lncRNAs may sponge miRNAs to attenuate their roles in both pathological and physiological processes without affecting their expression and accumulation [133]. Therefore, PLAC2 may regulate miR-19a/TNF-α to participate in PCOS.

The results of another human study revealed that miR-21 stimulation induces TLR8 expression and a subsequent immune response. This response was significantly increased in PCOS granulosa cells compared with normal GCs, where TLR8 overexpression corresponded with increased secretion of interferon-gamma (IFN-γ), TNF-α, and IL-12 [134]. The fact that miR-21/TLR8 signaling is involved in the regulation of inflammation, cell apoptosis and cell proliferation of PCOS granulosa cells provides profound new insight into the pathogenesis of PCOS.

## 4. Epigenetic Regulation of Insulin Resistance and Inflammatory Signaling in PCOS

Hyperinsulinemia caused by IR and reflecting impaired insulin action may potentiate the inflammatory response, as inflammatory pathways are linked to insulin signaling [135]. Briefly, the insulin/insulin receptor signaling cascade consists of two main branches emanating from the insulin receptor-insulin receptor substrate (IRS) node: the phosphatidylinositol 3-kinase (PI3K, a lipid kinase)/AKT (also known as PKB or protein kinase B) pathway and the Raf/Ras/MEK/MAPK (mitogen activated protein kinase, also known as ERK or extracellular signal regulated kinase) pathway [136,137,138]. After phosphorylation on tyrosine residues in their C-termini, IRS adaptor proteins serve as docking sites for the recruitment of downstream signaling effectors [139,140]. What may be relevant in PCOS is that the coexistence of inflammatory signaling via the NF-κB and activator protein 1 (AP-1)/c-Fos/c-Jun pathways results in the activation of serine kinases, I-kappa-B-kinase beta (IKKβ) and c-Jun N-terminal kinase 1 (JNK1), with a subsequent reduction in IRS signaling ability. In addition, induction of NO and a specialized family of proteins called suppressors of cytokine signaling (SOCS) leads to accelerated IRS degradation. NO also reduces PI3K/AKT activity by s-nitrosylation of AKT [141,142,143]. Finally, increased inflammatory gene expression even in the low-grade inflammatory response may negatively affect interactions with intracellular signaling components from nutrients and cytokines involved in the control of cell metabolism [144].

Considering that resistance to insulin action and—in the vast majority of cases—hyperandrogenism cause metabolic disorders typically observed in PCOS, which can be limited to some extent by diet and physical exercise, the influence of epigenetic mechanisms on the pathomechanism of PCOS is indisputable [145,146,147]. IR is inherently combined with low-grade inflammation in PCOS because a chronic inflammatory background creates an environment that predisposes patients to IR [14,148]. One can even speak of the epigenetic synergies between insulin resistance and inflammation, especially in the case of significant obesity [149,150,151].

The biological actions of insulin are mediated through a cell-surface receptor, the insulin receptor, which is present on the surfaces of virtually all mammalian cells, with particular emphasis on myocytes, hepatocytes, and adipocytes [152,153]. Therefore, the changes in the epigenome that accompany IR show similarities between the relevant tissues, i.e., muscle, liver, and adipose tissue [154,155]. The most widely studied epigenetic modification in IR is DNA methylation, including both global and targeted association studies. The latter made it possible to select the candidate genes. Hypermethylated or hypomethylated genes affecting IR and reported in tissues from patients with PCOS are listed in Table 1.

Interestingly, the colocation of histone deacetylases (HDACs) and insulin signaling components provides a rationale for explaining why intermittent fasting or short-term moderate caloric restriction may be an effective nonmedicinal treatment option for type 2 diabetes [190,191,192]. This is because activation of the members of the HDAC family, sirtuins, ameliorates inflammation by silencing the respective genes and promotes insulin sensitivity in adipocytes via PI3/AKT/mTOR and their downstream pathways [193,194].

In addition to disrupted methylation or acetylation of the epigenome in PCOS, the roles of miRNAs have recently attracted considerable attention in the context of IR and inflammation. For example, miR-146a, miR-155, and miR-486 were upregulated, whereas miR-320 and miR-370 were downregulated in GCs from PCOS patients. Comparative studies (PCOS vs. PCOS-free controls) of miR-146a, miR-155, and miR-486 in follicular fluid revealed lowered levels in PCOS [195]. These miRNAs showed relationships with BMI, IR, E, the number of dominant follicles, and high mobility group box-1 (HMGB-1). It is worth noting that HMGB-1 is considered an essential facilitator/inducer of local and systemic inflammatory responses [195,196]. Another miRNA considered in the context of the PCOS pathomechanism, miR-29c-3p, shows opposite effects with respect to inflammation, depending on the tissue and signaling pathways used. Hence, miR-29c-3p may promote intestinal inflammation by targeting leukemia inhibitory factor (LIF) in ulcerative colitis and, through targeting the transcription factor forkhead box protein O3 (FOXO3A), may ameliorate inflammatory damage of cells by NOD-, LRR- and pyrin domain-containing protein 3 (NLRP3) inflammasome inhibition [197,198,199]. Moreover, miR-29c-3p participates in insulin function by targeting FOXO3A. Hyperinsulinemia and IR in PCOS were positively correlated with downregulation of miR-29 c-3p and upregulation of FOXO3A [200]. Another PCOS study found that the network of miR-106-5p/miR-155-5p targets was mostly concentrated in pathways related to IR and inflammation, with the upregulation of the inflammatory genes Il18/IL18 and Socs3/SOCS3 [201].

The involvement of lncRNAs in IR and the inflammatory background in PCOS is the subject of ongoing research; however, in diabetes, which shows many metabolic similarities to PCOS, clinically relevant evidence for the association of altered lncRNAs with poor glycemic control, IR, accelerated cellular senescence, and inflammation has been proven [202].

## 5. Epigenetic Modulation of Androgenic Activity in Response to Inflammatory Signaling in PCOS

It should be mentioned that androgens exert several inhibitory effects on immune cell activity with predominantly anti-inflammatory properties [203,204]. In concordance, emerging evidence from independent scientific studies indicates that the activity of the immune system is superior in females and that androgens, found in much higher concentrations in males, may act as “immunosuppressive” molecules with inhibitory effects on inflammatory reactions [205]. For example, testosterone may inhibit inflammatory reactions such as the expression of Toll-like receptor 4 (TLR4) on neutrophils, secretion of interferon γ (IFNγ) from natural killer (NK) cells, and/or the biosynthesis of TNFα and iNOS-derived NO in macrophages [205]. Moreover, androgens promote anti-inflammatory conditions, including increased production of transforming growth factor β (TGFβ) and IL-10, as well as overexpression of peroxisome proliferator-activated receptor (PPAR)α in T cells [206,207]. It is no coincidence that asthma, rheumatoid arthritis, allergic rhinitis, systemic lupus erythematosus (SLE) and many other disorders related to an altered immune response have a higher incidence and severity in women than in men [205,208].

Taking the above into account, most researchers express the view that—in the pathophysiology of PCOS—inflammatory stimuli (i.e., low-grade inflammation) trigger increased androgen production and shifts in gene expression in thecal-interstitial cells [209]. An unresolved issue that requires further research remains to clarify whether inflammation-induced hyperandrogenism is a manifestation of a compensatory mechanism to restore homeostasis within the immune system. Hyperandrogenism present in PCOS, in turn, promotes a proinflammatory TNFα response to glucose administration. It was demonstrated that androgen excess increases androgen receptor (AR) mRNA content within mononuclear cells (MNCs) and TNFα release from MNCs in an AR-dependent fashion [210]. Both follicular excess of androgens and increased levels of insulin downregulate aromatase in luteinized granulosa cells in PCOS women, contributing to the mechanism of a vicious cycle between inflammation, IR and hyperandrogenism [211,212].

Data on the influence of the inflammatory response on the production of androgens as a result of posttranscriptional regulation of gene expression in PCOS are being constantly updated [209,213,214]. Thus, any epigenetic regulation that leads to a decrease in the expression of the CYP19A1 (aromatase) gene in granulosa cells or a decrease in the relative expression rate of GC aromatase, as well as producing upregulation of AR and increased activity of the key enzymes essential for androgen biosynthesis in thecal cells (i.e., CYP11A1, CYP17, and HSD3B2), should be considered in the context of inflammation in PCOS [211,215,216,217] (Figure 3).

For example, as mentioned elsewhere, increased expression levels of miR-9, miR-18b, and miR-135a in follicular fluid are correlated with enhanced synthesis of steroids, whereas downregulated expression of miR-130b-3p in thecal cells results in excessive androgen biosynthesis [69,70]. Differentially expressed in PCOS domain containing 1A (DENND1A), especially DENND1A variant 2 (DENND1A.V2), was identified as a target site for miR-130b-3p activity. The comparison of miR-130b-3p in normal and PCOS thecal cells demonstrated decreased miR-130b-3p expression in PCOS thecal cells, which was correlated with increased DENND1A.V2, cytochrome P450 17α-hydroxylase (CYP17A1) mRNA and androgen biosynthesis [70]. Additionally, miR-130b-3p is highly expressed in PCOS and promotes GC proliferation by targeting SMAD family member 4 (SMAD4). This may be important because the bone morphogenetic protein 4 (BMP4)-SMAD signaling pathway regulates the estrogen/androgen ratio. BMP4 treatment suppressed androgen synthesis in thecal cells and promoted estrogen production in granulosa cells by regulating the expression of steroidogenic enzymes, including CYP11A, HSD3B2, CYP17A1, and CYP19A1. BMP4 levels are significantly decreased in hyperandrogenism [218,219]. The role of miR-29a in the regulation of the inflammatory background via AR needs clarification. A significant downregulation of miR-29a in PCOS patients was associated with an increased antral follicle count and decreased aromatase expression and estradiol production in GCs. However, miR-29a downregulation may exert both anti-inflammatory and proinflammatory effects, as was demonstrated in endothelial and hepatic cells, respectively [220,221].

## 6. Concluding Remarks

Low-grade inflammation creates a constant background disease in PCOS [13,14,15]. Several inflammation- and stress-related signaling pathways have been identified as key players in these processes, including the nuclear factor-κB, signal transducer and activator of transcription (STAT) and stress- and mitogen-activated protein kinase (MAPK) pathways [222,223]. The functioning of many other signaling pathways, not directly related to inflammatory signal propagation, can be altered under these proinflammatory conditions [224]. While the etiopathogenesis of PCOS is still not adequately understood, environmental factors are consistently mentioned as key in the development of the disease [7,34,35,36]. Thus, epigenetic factors should also be considered with regard to their therapeutic potential [225,226,227]. The possibility of influencing the processes related to oxidative stress and ROS formation, mitochondrial dysfunction with the release of DAMPs into the cytoplasm, and reversal of the proinflammatory cytokine profile toward normal and elimination of IR may be of high significance [37,39,135]. Moreover, effective treatment of the inflammatory background in PCOS can initiate normalization in androgenic activity, thereby removing the major endocrine disorder that limits a patient’s ability to become pregnant [148,228]. Despite the clear link between inflammation and IR at the level of epigenetic regulation, both processes should be treated separately due to the definition of therapeutic goals. Anti-inflammatory treatment should improve the effectiveness of insulin action by reduction of IR, whereas inhibition of the epigenetic factors that promote IR can reduce inflammatory background. An important problem in conducting this type of research is lack of a good animal model for human PCOS [229,230].

## Figures and Tables

**Figure 1 ijms-23-14663-f001:**
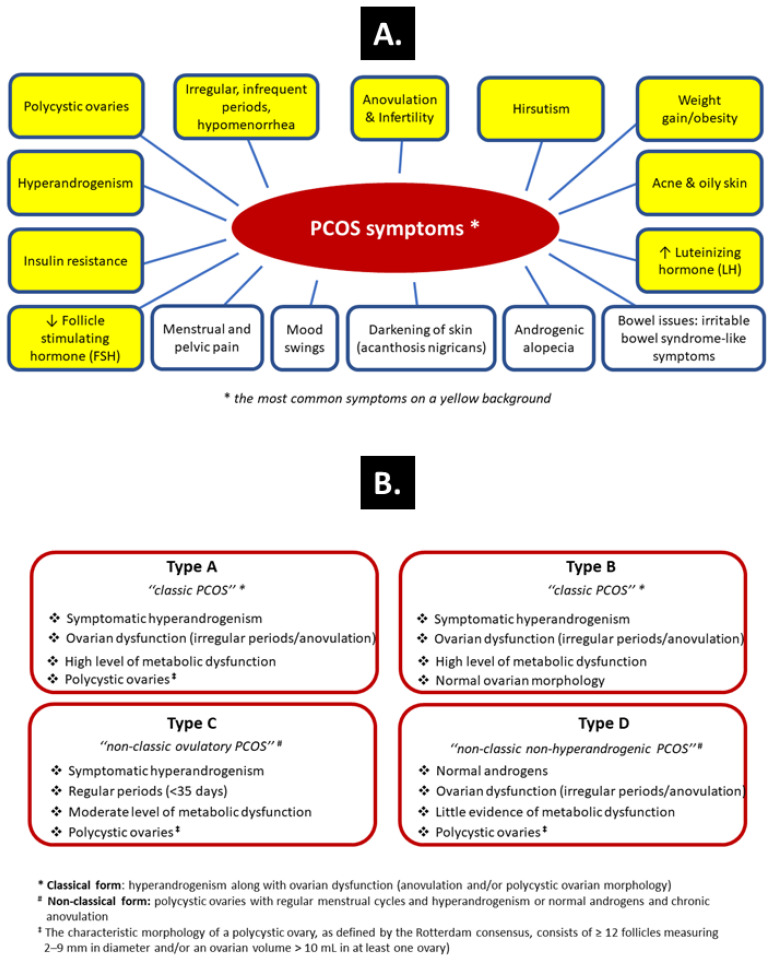
Symptoms of PCOS (**A**) and the simple clinical classification taking into account the phenotypes of PCOS (**B**) [6].

**Figure 2 ijms-23-14663-f002:**
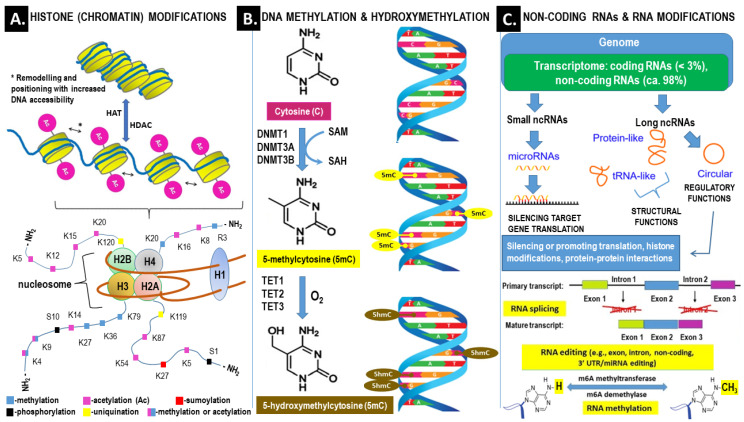
Overview of the main epigenetic mechanisms that regulate gene expression and may establish potentially heritable changes in gene expression without altering the underlying DNA nucleotide sequence. (**A**) Histone (chromatin) modifications. At the top. Chromatin remodeling is the dynamic modification of chromatin architecture to allow access of condensed genomic DNA to the regulatory transcription machinery proteins, and thereby control gene expression. For example, histone acetylation by HAT (histone acetyl transferase) increases DNA (chromatin) accessibility because acetylated histones cannot pack as well together as deacetylated histones. HDAC–histone deacetylase; At the bottom. Each nucleosome consists of two subunits, both made of histones H2A, H2B, H3 and H4, also known as core histones, with the linker histone H1 acting as a stabilizer. Histone post-translational modifications are covalent modifications of histones by phosphorylation on serine or threonine residues, methylation on lysine or arginine, acetylation and deacetylation of lysines, ubiquitylation of lysines and sumoylation of lysines. Histone modifications affect chromosome structure and function, especially during transcription and chromatin remodeling processes. (**B**) DNA methylation and hydroxymethylation. DNA can be modified at cytosine and adenine residues by the addition of chemical groups. Cytosines can be modified by methylation (5mc) or hydroxymethylation (5hmC), while adenines are modified by methylation. CpG islands (regions of the genome that contain a large numer of CpG dunucleotide repeats) are DNA methylations regions in promoters known to regulate gene expression through transcriptional silencing of the corresponding gene. DNA methylation at CpG islands is crucial for gene expression and tissue-specific processes. *DNMT–DNA methyltransferase*; *SAM–S-adenosylmethionine*; *SAH–S-adenosylhomo-cysteine*; *TET–ten-eleven-translocation* (*methylcytosine dioxygenase*). (**C**) Non-coding RNAs (ncRNAs) and RNA modifications–ncRNAs play an important role in transcription regulation by epigenetic machinery. Within RNA-induced silencing complexes (RISCs), miRNAs mediate the recognition and binding of RNAs that become targeted for degradation. lncRNAs are associated with other complexes and can activate or repress transcription-Alternative splicing (AS) of pre–mRNAs serves as an additional regulatory process for gene expression after transcription, and it generates distinct mRNA species, and even noncoding RNAs (ncRNAs), from one primary transcript. AS contributes to the diversity of proteins in eukaryotes as cells respond to signals from the environment. AS may lead to generation of ncRNAs, especially long noncoding RNAs (lncRNAs). RNA modifications, such as the RNA N6-methyladenosine (m6A) modification, have been found to regulate AS–RNA editing is an important mechanism of genetic regulation that amplifies genetic plasticity by allowing the production of alternative protein products from a single gene. RNA editing involves the post-transcriptional insertion and deletion of nucleotides (e.g., uridylate–UMP) within nascent transcripts. RNA editing has been observed in mRNAs, tRNAs, and rRNAs, in mitochondrial and chloroplast encoded RNAs, as well as in nuclear encoded RNAs–RNA methylaton is a post-transcriptional level of regulation. At present, more than 150 kinds of RNA modifications have been identified. They are widely distributed in messenger RNA (mRNA), transfer RNA (tRNA), ribosomal RNA (rRNA), noncoding small RNA (sncRNA) and long-chain non-coding RNA (lncRNA).

**Table 1 ijms-23-14663-t001:** Hypermethylated or hypomethylated genes affecting IR including those found in tissue from patients with PCOS (marked in red). Modified from the respective review papers [44,155].

Tissue	Exemplary Loci with Differential Methylation	References
Peripheral blood	*LY6G6F*, *KCTD21*, *ADCY9*, *RABL2B*, *ZNF611*, *VASH1*, *FST*, *LMNA*, *PPARGC1A*, *L-1*, *TMSB15B*, *RPF1*, *DNA2*, *EPHA8*, *LHCGR*, *EPHX1*, *JAML*, *KBTBD12*, *SLC29A1*, *GPR176*, *MYOZ2*, *PIGT*, *C2CD4B*, *PCDHA7*, *HMGA1*, *PCDH18*	Sang et al., 2013, 2014 [156,157]; Shen et al., 2013 [158]; Ting et al., 2013 [159]; Wang et al., 2014 [160]; Li et al., 2017 [161]; Sagvekar et al., 2017 [162]; Zhao et al., 2017 [163]
	*CLCA4*, *LECT1*, *CXCR1*, *HDAC4*, *IGFR1*, *LEPR*, *ABCG1*, *SH3RF3*, *MAN2C1*	Arpon et al., 2019 [164]
	*SLC19A1*, *EFNA2*	Day et al., 2017 [165]
	*LETM1*, *RBM20*, *IRS2*, *MAN2A2*, *1q25.3*, *FCRL6*, *SLAMF1*, *APOBEC3H*, *15q26.1*	Liu et al., 2019 [166]
	*ABCG1*, *PHOSPHO1*, *SOCS3*, *SREBF1*, *TXNIP*	Chambers et al., 2015 [167]
	*ABCG1*, *PHOSPHO1*	Dayeh et al., 2016 [168]
	*ABCG1*, *PHOSPHO1*, *SREBF*, *NFATC2IP*, *KLHL18*, *FTH1P20*	Wahl et al., 2017 [169]
	*ABCG1*, *SREBF1*, *TXNIP*, *PROC*, *SLC43A1*, *PHGDH*, *MAN2A2*	Cardona et al., 2019 [170]
	*Alu element repeats methylation*	Zhao et al., 2012 [171]
	*SLC7A11*, *SLC1A5*, *SLC43A1*, *PHGDH*, *PSORS1C1*, *SREBF1*, *ABCG1*	Ma et al., 2019 [172]
	*SLC7A11*, *SLC43A1*, *SLC1A5*, *PHGDH*, *PSORS1C1*, *SREBF1*, *ANKS3*	Nano et al., 2017 [173]
Umbilical cord blood	* PRKN * , *PAX6*, *B4GALT7*, *MEST*, *CACNA2D2*, *RGMA*, *PRDM10*, * ESR1 * , *APP*, *RBPMS LHCGR*, *CASP10*, *SPHK1*, *PCSK6*, *ARHGAP45*, *MIB2*	Lambertini et al., 2017 [174]
Whole ovarian tissue	* FBN1 * , *NAV2*, *PRDM1*, *RNF213*, *SSBP2*, *TNIK*, *ZFAND3*, *ZNF503*, *SLC2A8*, *NRIP1*, *IGF2BP2*, *CYP19A1*, *AMHR2*, *SNURF*, *SUMO3*, *PNMA6A*, *ADRA1D*, *SCML1*, *C2CD6*, *NR0B1*, *INSR*, *AMH*, *SPANXD*, *TUBA3E*, *FAM47B*, *MAB21L1*, *RBM3*	Yu et al., 2013, 2015 [175,176]; Wang et al., 2014 [177]
Granulosa cells	* MATN4 * , *DLGAP2*, *CDH13*, *GAREM2*, *GSC*, *ANKRD34C*, *ATP8B2 and PPARG*, *L-1*, *LHCGR*, *SMG6*, *CCR5*, *LHB*, *NTN1*, *ARFGAP1*, *MDGA1*, *NCOR1*, *YAP1*, *CD9*, *NR4A1*, *EDN2*, *BNIP3*, *LIF*	Qu et al., 2012 [178]; Wang et al., 2014 [160]; Xu et al., 2016 [179]; Jiang et al., 2017 [180]; Sagvekar et al., 2017 [162]
Liver	*GRB10*, *PPP1R1A*, *IGFBP2*, *ABCC3*, *MOGAT1*, *PRDM16*	Nilsson et al., 2015 [181]
	*PRKCE*, *PDGFA*	Kirchner et al., 2016 [182]
	*PDGFA*	Abderrahmani et al., 2018 [183]
	*SYT7*, *LTBR*, *CATSPER2*, *LPAL2*, *NCALD*, *ZDHHC11*, *LGTN*, *OXT*, *PRSS21*	Barajas-Olmos et al., 2018 [154]
	*IGFBP2*, *IGF1*, *PRKCE*, *PGC1A*, *SREBF2*, *FOXA1*, *FOXA2*, *ZNF274*	Hotta et al., 2018 [184]
	*FGFR2*, *MAT1A*, *CASP1*, *COL1A1*, *COL1A2*, *COL4A1*, *COL4A2*, *LAMA4*, *LAMB1*, *CTGF*, *PDGFA*, *CCR7*, *CCL5*, *STAT1*, *TNFAIP8*	Murphy et al., 2013 [185]
	*PPARGC1A*, *DNMT1*, *HDAC9*, *ALKBH5*, *LDHB*, *COL4A1*, *ARL4C*, *SEMA3E*, *ITGB4*	de Mello et al., 2017 [186]
	*E2F1*, *TFAP2A NFKB1*, *HNF4A*, *HNF1A*, *SREBF1*, *TCF4*, *ETS1*	Bysani et al., 2017 [187]
	*FGFR2*, *IGF1*, *MTHFD2*, *PTGFRN*, *ZBTB38*, *MGMT*, *FBLIM1*, *CYR61*, *NQO1*	Hotta et al., 2018 [184]
Subcutaneous adipose tissue	* ZZEF1 * , *TPT1*, *STUB1*, *DMAP1*, *RAB5B*, *PPARG*, *SVEP1*, *SAV1*, *RORA*, *RAB6A*, *CNST*, *PUM1*, *DIP2C*, *SNX8*, *SRGAP3*, *ZFHX3*, *OR52W1*, *BBX*	Kokosar et al., 2016 [188]
Skeletal muscle	* CST3 * , *SPRTN*, *COL1A1*, *SCMH1*, *VAT1*, *CSPP1*, *ERP29*, *ADK*, *KLF10*, * HEATR3 * , *HJV*, *MAP2K6*, *FOXO3*	Nilsson et al., 2018 [189]

## Data Availability

Not applicable. This review is based on already published data listed in the references.

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
