# Peer review of "Modulation of the Inflammatory Response in Polycystic Ovary Syndrome (PCOS)—Searching for Epigenetic Factors"

_ijms, 2022, doi:10.3390/ijms232314663_

Round 1

Reviewer 1 Report

I am pleased to revise this manuscript regarding the impact of the inflammatory milieu associated with PCOS. 

I found the paper very intriguing and interesting and, thus, in my opinion, it could be suitable for publication. 

Anyway, I suggest to add the following references to the paper in order to enhance its scientific strength:

- To improve the section "introduction": Gill V, Kumar V, Singh K, Kumar A, Kim JJ. Advanced Glycation End Products (AGEs) May Be a Striking Link Between Modern Diet and Health. Biomolecules. 2019 Dec 17;9(12):888. doi: 10.3390/biom9120888. PMID: 31861217; PMCID: PMC6995512.

- To mention the major treatments involved in the regulation of pro-inflammatory pattern of PCOS:

Xue J, Li X, Liu P, Li K, Sha L, Yang X, Zhu L, Wang Z, Dong Y, Zhang L, Lei H, Zhang X, Dong X, Wang H. Inulin and metformin ameliorate polycystic ovary syndrome via anti-inflammation and modulating gut microbiota in mice. Endocr J. 2019 Oct 28;66(10):859-870. doi: 10.1507/endocrj.EJ18-0567. Epub 2019 Jul 3. PMID: 31270279; 

Duleba AJ. Medical management of metabolic dysfunction in PCOS. Steroids. 2012 Mar 10;77(4):306-11. doi: 10.1016/j.steroids.2011.11.014. Epub 2011 Dec 13. PMID: 22182833; PMCID: PMC3409585.

Fruzzetti F, Capozzi A, Canu A, Lello S. Treatment with d-chiro-inositol and alpha lipoic acid in the management of polycystic ovary syndrome. Gynecol Endocrinol. 2019 Jun;35(6):506-510. doi: 10.1080/09513590.2018.1540573. Epub 2019 Jan 7. PMID: 30612488.

Additionally, a specific section concerning "Materials and Methods" should be advisable.

Sincerely

Author Response

Thank you very much for taking the time to prepare your review and substantive comments to improve the final version of the publication.

Please find below our point-by-point response:

COMMENT: English language and style are fine/minor spell check required

RESPONSE: As usual, the final text of the review was linguistically verified by the native speaker scientific experts from American Journal Experts (AJE). The Certificate from AJE is included. You can learn from it that the text “was edited for proper English language, grammar, punctuation, spelling, and overall style by one or more of the highly qualified native English speaking editors at AJE.” When using of this service provider (AJE)  many times, I have never come across any comments from reviewers regarding language errors or incomprehensible style.

COMMENT: I suggest to add the following references to the paper in order to enhance its scientific strength:

- To improve the section "introduction": Gill V, Kumar V, Singh K, Kumar A, Kim JJ. Advanced Glycation End Products (AGEs) May Be a Striking Link Between Modern Diet and Health. Biomolecules. 2019 Dec 17;9(12):888. doi: 10.3390/biom9120888. PMID: 31861217; PMCID: PMC6995512.

- To mention the major treatments involved in the regulation of pro-inflammatory pattern of PCOS:

Xue J, Li X, Liu P, Li K, Sha L, Yang X, Zhu L, Wang Z, Dong Y, Zhang L, Lei H, Zhang X, Dong X, Wang H. Inulin and metformin ameliorate polycystic ovary syndrome via anti-inflammation and modulating gut microbiota in mice. Endocr J. 2019 Oct 28;66(10):859-870. doi: 10.1507/endocrj.EJ18-0567. Epub 2019 Jul 3. PMID: 31270279;

Duleba AJ. Medical management of metabolic dysfunction in PCOS. Steroids. 2012 Mar 10;77(4):306-11. doi: 10.1016/j.steroids.2011.11.014. Epub 2011 Dec 13. PMID: 22182833; PMCID: PMC3409585.

Fruzzetti F, Capozzi A, Canu A, Lello S. Treatment with d-chiro-inositol and alpha lipoic acid in the management of polycystic ovary syndrome. Gynecol Endocrinol. 2019 Jun;35(6):506-510. doi: 10.1080/09513590.2018.1540573. Epub 2019 Jan 7. PMID: 30612488.

RESPONSE: The references listed have been added to the corrected version of the manuscript. Appropriate changes to the numbering of citations have also been made.

COMMENT: Additionally, a specific section concerning "Materials and Methods" should be advisable.

RESPONSE: The subsection 1.1. Methodology of literature searching has been added.

Reviewer 2 Report

I have read with great interests the article entitled “Modulation of the inflammatory response in polycystic ovary syndrome (PCOS) – searching for epigenetic factors” by Dariusz Szukiewicz, Seweryn Trojanowski, Anna Kociszewska, and Grzegorz Szewczyk. In their work, the authors focus on collecting evidence on the inflammatory processes that may influence the epigenetic regulation of genes and thus alter the gene expression in women with PCOS. Indeed, they point out both mechanisms that arise from PCOS and mechanisms that may cause PCOS onset. Their work is really valid and consistent, and thus I believe that it will be suitable for publication after a minor revision of the English (e.g., in the first paragraph, there is a list of symptoms derived from androgen excess in bracket that is not very clear (“anovulation, polycystic ovary morphology, lack of or scanty, irregular menstrual periods, acne and hirsutism”)).

Author Response

Thank you very much for taking the time to prepare your review and substantive comments to improve the final version of the publication.

Please find below our point-by-point response:

COMMENT: English language and style are fine/minor spell check required

RESPONSE: As usual, the final text of the review was linguistically verified by the native speaker scientific experts from American Journal Experts (AJE). The Certificate from AJE is included. You can learn from it that the text “was edited for proper English language, grammar, punctuation, spelling, and overall style by one or more of the highly qualified native English speaking editors at AJE.” When using of this service provider (AJE)  many times, I have never come across any comments from reviewers regarding language errors or incomprehensible style.

COMMENT: ….. in the first paragraph, there is a list of symptoms derived from androgen excess in bracket that is not very clear (“anovulation, polycystic ovary morphology, lack of or scanty, irregular menstrual periods, acne and hirsutism”)).

RESPONSE: In order to increase readability, the cited fragment of the text has been rephrased. The revised version of the manuscript now reads as follows:

Primary clinical manifestations of PCOS are derived from the excess of androgens and include absence of ovulation (anovulation), polycystic ovarian morphology, lack of or scanty, irregular menstrual bleedings, acne – an inflammatory condition of the  skin in which the skin's sebaceous glands become clogged and infected, and excessive growth of dark or coarse hair in a male-like pattern – face, chest and back (hirsutism). The secondary manifestations include multiple metabolic, cardiovascular, and psychological disorders (Fig. 1A).
